# The Phytopathogen *Fusarium verticillioides* Modifies the Intestinal Morphology of the Sugarcane Borer

**DOI:** 10.3390/pathogens12030443

**Published:** 2023-03-11

**Authors:** Diego Z. Gallan, Maressa O. Henrique, Marcio C. Silva-Filho

**Affiliations:** Departamento de Genética, Escola Superior de Agricultura Luiz de Queiroz, Universidade de São Paulo, Piracicaba 13418-900, SP, Brazil; diego.gallan@usp.br (D.Z.G.); maressahenrique@usp.br (M.O.H.)

**Keywords:** colonization of microvilli, *Diatraea saccharalis*, insect–fungus interactions, red rot disease

## Abstract

Background: In tropical sugarcane crops, the fungus *Fusarium verticillioides*, the agent responsible for the occurrence of the red rot complex, occurs in association with the sugarcane borer *Diatraea saccharalis*. This fungus, in addition to being transmitted vertically, can manipulate both the insect and the plant for its own dissemination in the field. Due to the complex interaction between *F. verticillioides* and *D. saccharalis*, and the high incidence of the fungus in the intestinal region, our objective was to investigate whether *F. verticillioides* could alter the intestinal structure of the insect. Methods: We combined analysis of scanning electron microscopy and light microscopy to identify whether the presence of the fungus *F. verticillioides*, in artificial diets or in sugarcane, could lead to any alteration or regional preference in the insect’s intestinal ultrastructure over the course of its development, or its offspring development, analyzing the wall and microvillous structures of the mid-digestive system. Results: Here, we show that the fungus *F. verticillioides* alters the intestinal morphology of *D. saccharalis*, promoting an increase of up to 3.3 times in the thickness of the midgut compared to the control. We also observed that the phytopathogen colonizes the intestinal microvilli for reproduction, suggesting that this region can be considered the gateway of the fungus to the insect’s reproductive organs. In addition, the colonization of this region promoted the elongation of microvillous structures by up to 180% compared to the control, leading to an increase in the area used for colonization. We also used the fungus *Colletotrichum falcatum* in the tests, and it did not differ from the control in any test, showing that this interaction is specific between *D. saccharalis* and *F. verticillioides*. Conclusions: The phytopathogenic host *F. verticillioides* alters the intestinal morphology of the vector insect in favor of its colonization.

## 1. Introduction

Interactions involving plant–insect pathogens are influenced by many shared complex evolutionary properties, effects of external organisms, environmental conditions, defense genes and effectors, the understanding of which can support the discovery of new mechanisms and strategies for the management of insect pests in a wide range of agroecosystems [1,2,3,4]. Due to the high complexity, most studies on plant interactions with pathogens and herbivore insects have been focused on binary interactions; however, a holistic view of multiple interactions has gained more attention because of the direct or indirect effects of one organism on the other [5]. Although studies of interactions are highly complex and recent, it has already become possible to identify many insect–microorganism associations [6], as in the case of observations of the absorption of nutrients from the host [7,8,9]. In addition, this interaction might be an excellent and efficient machine for microorganism dispersion [10,11,12] through the use of different propagation strategies, such as the release of volatile organic compounds (VOCs) [10,13,14,15], enzymatic activation modulating gustatory sensors that alter taste [16], and even direct action on the nervous system within the head capsule in the extraordinary study of “zombie” ants [17].

The manipulation that microorganisms used to be transmitted and disseminated by insects highlights the complexity of these multiple interactions and involves numerous strategies, including establishment in the host organism [18,19,20,21]. A wide range of microorganisms is found in intestines [10,22], body cavities [23], hemolymph [24,25], and cells [26], and also attached to the cuticle [27], among other structures, in a wide variety of insects [28,29]. One of the most well-studied interactions is related to insects and bacteria living in the midgut [30], as the intestine is rich in nutrients, thus favoring installation and colonization by microorganisms. In general, bacteria are located inside specialized cells attached to the intestinal membranes or are found free in the intestinal region [30,31,32]. There are few studies involving the role of the intestine in fungal–insect interactions. Recently, a mutualism scenario has been proposed between *Penicillium* and *Bactrocera dorsalis* (oriental fruit fly, Hendel), in which the host colonizes the insect intestine providing the ideal nutritional balance for development, and in turn the insect promotes the spread of the fungus [33].

Another complex symbiotic interaction between a fungal phytopathogen and an herbivorous insect was discovered, and it was shown that the fungal phytopathogen manipulates both the plant and insect responses to promote its dissemination. Additionally, the fungi were found to be vertically transmitted to the insect offspring [10].

In Brazil, infection by the sugarcane borer *Diatraea saccharalis* is usually associated with the fungal pathogens *Fusarium verticillioides* and *Colletotrichum falcatum*, which form the borer-rot complex [5,34,35]. Due to the complex interaction between *F. verticillioides* and *D. saccharalis,* we aimed to investigate whether *F. verticillioides* might alter the structure of insect intestines. The initial steps of fungal colonization in insect intestines may provide evidence of the mechanism by which the fungus moves from the gut to the insect reproductive organs, ensuring vertical transmission.

## 2. Materials and Methods

### 2.1. Fungal Culture

Isolates of *F. verticillioides* and *C. falcatum* were collected from sugarcane plants and cultivated in potato dextrose (PD) medium (Difco, Sparks, NV, USA) maintained at 26 °C with a 12 h photoperiod.

### 2.2. Insect Rearing

*D. saccharalis* was provided by Prof. Dr. José RP Parra from the University of São Paulo, Piracicaba. The larvae were fed an artificial diet [36] and kept in a room under controlled conditions (temperature 25 ± 3 °C, relative humidity 60 ± 10% and 12 h of light). Adults were kept in cages covered with white paper sheets, where the eggs were deposited, collected and cleaned with a 1% copper sulfate solution daily. Newly hatched caterpillars were transferred to the artificial diet.

### 2.3. Sugarcane Cultivation

Presprouted clonal seedlings (PSS) of sugarcane (*Saccharum* spp. cv. ’RB975952’) were acquired from Explante Biotecnologia based in Mogi Mirim-SP. PSS were cultivated in 6 L plastic pots filled with substrate (Fert Solo, Limeira, SP, Brazil) and fertilized every two weeks (Plant-Prod 20-20-20, Master Plant-Prod, Brampton, ON, Canada) according to the recommendations from the manufacturer. The plants were kept in an insect-free greenhouse under natural light and natural variations in temperature and humidity.

### 2.4. Artificial Diet Contaminated with F. verticillioides or C. falcatum

Seven days before the experiment, a total of 5 × 10^4^ fungal conidia of *F. verticillioides* or *C. falcatum* were inoculated into a Petri dish containing 10 mL of sterile diet, using 50 μL of ddH_2_O as the negative control. Nipagin and formaldehyde were removed from all diets used.

### 2.5. Plants Infected with F. verticillioides or C. falcatum

Forty-day-old plants were artificially inoculated with 50 μL of a suspension of *F. verticillioides* or *C. falcatum* at a concentration of 1 × 10^6^ conidia/mL using a sterile syringe. Subsequently, the wound was covered with colored plastic adhesive to prevent the entry of other pathogens and to identify the region of inoculation. The plants were grown for another seven days in a glasshouse, with temperature and relative humidity being monitored at every stage after inoculation (Figure 1), using a thermo-hygrometer (MX1101, OnSet, United States of America). Only plants that showed symptoms were used in the experiments. The same procedure was performed with a simulated, noninoculated plant, except that the solution did not contain conidia. Compared to healthy plants, the induction treatment (mock) did not change the behavior of *D. saccharalis*.

### 2.6. Contamination of Insects by Fungi

For tests on artificial diets, groups of ten third-instar larvae (P generation) of *D. saccharalis* were placed in the center of Petri dishes containing a diet colonized with *F. verticillioides* or *C. falcatum* and in plates with a sterile diet as a negative control. The respective diets were changed every two days to avoid contamination. This process was carried out until the larvae reached the pupa stage, where the insects were removed from the diet, sexed and sterilized for two minutes in a 1% sodium hypochlorite solution and then placed in distilled water for one minute to remove remnant adhering materials and leftover products. Afterward, three mating pairs were placed in 300 mL plastic cups containing a cotton ball soaked in sterilized water, which remained until they achieved adulthood. After emergence of the adults, sterile white paper was added to the walls of the cups for egg laying. The papers were removed daily, and the egg deposits were collected, cleaned with a 1% copper sulfate solution and inserted (±20 eggs) in the center of a Petri dish containing an artificial diet without anticontaminants.

After hatching (F1 generation), the larvae were submitted to the same procedure, with collections of fifth-instar larvae, mature pupae and newly emerged adults for microscopy tests. The entire process was carried out with sterile material under a hood to avoid contamination and stored in a room under controlled conditions.

In the tests using sugarcane, one day before the experiments, the infected and healthy plants were transferred to the laboratory and kept under controlled conditions. After the third instar, the larvae (P generation) were inserted individually into transparent and sterile Eppendorf microtubes^®^ of 0.5 mL and affixed perpendicularly in the second internode of the sugarcane with the aid of transparent adhesive tape for fixation. After 15 days, with the aid of pruning pliers, cuts were made at the ends of the stem and taken to the laboratory, where small cuts were made to locate and collect the pupae, which were sexed, sterilized and submitted to the procedure previously presented until egg deposits were obtained.

Utilizing the egg deposits of the P generation, healthy 25-day-old plants were transferred to the laboratory and concurrently ± three eggs (F1 generation) attached in transparent and sterile 1.5 mL Eppendorf^®^ microtubes. The microtubes were placed perpendicularly in the middle region of the stem with the aid of adhesive tape, and the beginning of hatching was observed daily.

In a period ranging from 12 to 15 days, fifth-instar larvae were collected from the interior of the stems for microscopy tests. Pupae were collected from 18 to 23 days from the plant, some of which were packaged to obtain and collect the adults. The environmental variations of the entire process were recorded (Figure 1). Prior to the beginning of the experiments, larvae were fasted for 24 h.

### 2.7. Assay with D. saccharalis Insects by Light Microscopy (LM)

Insects infected with *F. verticillioides* or *C. falcatum* or that were not contaminated were used. The insects were fixed in Karnovsky’s solution for five days [37]. During this period, the samples were placed in a vacuum pump at 8 °C for better penetration and preservation of the fixative. Subsequently, they were dehydrated using a graded series of alcohol (10, 20, 30, 50, 70, 80, 90 and 100%) and embedded in paraffin (Allkplast, Campinas, São Paulo, Brazil). The samples were sectioned using a Leica RM 2255 rotational microtome with a thickness of 8 μm. For histological analyses, part of the cross-sections were placed on microscope slides coated with a mixture of 1.5% ovalbumin and 3% glycerol in distilled water. Sections were deparaffinized in 100% toluene and then stained with hematoxylin–eosin [38]. Slides containing midgut structures were examined and micrographed with a digital microscope (KH 8700, Hirox; Tokyo, Japan).

### 2.8. Assay with Intestines of D. saccharalis by Scanning Electron Microscopy (SEM)

To obtain the intestines, uncontaminated insects infected solely with *F. verticillioides* or *C. falcatum* were placed in glass Petri dishes at −2 °C using forceps and a scalpel [10]. The intestines of *D. saccharalis* fifth-instar larvae, pupae and adults were removed and fixed in Karnovsky’s solution for 48 h [37] and in a vacuum pump at 8 °C for better penetration and preservation of the fixative. Afterward, they were transferred to a 30% glycerin solution for 1 h as a cryopreservative. Then, the samples were immersed in liquid nitrogen, cross-cracked, washed in deionized water, dehydrated in a series of acetone solutions (25, 50, 75, 90 and 100%) and dried to critical points (LEICA CPD 300) [39]. Finally, the samples were glued onto gold-plated aluminum stubs (Baltec model SCD 050) and examined under an SEM (Jeol JSM IT 300) at 20 kV.

### 2.9. Experimental Design for Quantification of Intestinal Thickness of D. saccharalis

Measurements of the thickness of the insect intestinal membrane and structures were made with the images captured by the microscopy equipment, so that three equidistant regions of each image were analyzed and the average was obtained, with the statistical analysis performed using the Tukey test, with significance levels set at *p* < 0.05 [40]. Ten insects were used for each test, and three images of each insect were analyzed. Light microscopy analyses were performed for insects from artificial diets and from sugarcane plants. For the SEM analyses, only insects from an artificial diet were used. Two independent experiments were performed.

### 2.10. Experimental Design for Analysis of Intestinal Microvilli of D. saccharalis

The microvillous structures of the intestines of *D. saccharalis* were analyzed with the images captured by SEM. Ten intestines of *D. saccharalis* of the F1 generation, from the P generation contaminated with *F. verticillioides* or *C. falcatum* or without contaminants, were used in the tests.

To obtain the length of the microvilli, ten equidistant microvilli from each intestine were analyzed and the average was obtained with the statistical analysis performed by the Tukey test, with significance levels set at *p* < 0.05.

Two independent experiments were performed.

## 3. Results

### 3.1. Effects of F. verticillioides on the Intestines of D. saccharalis

Our data showed that there was an enlargement of the intestinal wall of larvae, pupae and adult descendants of insects fed a diet colonized by *F. verticillioides* (Figure 2). The intestinal membrane of the larvae presented an average thickening of 2.7-fold in relation to the control (Figure 2a), whereas for the pupae, the increase was 2.58-fold (Figure 2c) and was 3.01-fold for adults (Figure 2e).

There was no significant difference between the trials using artificial diet or sugarcane plants, and the only differences were between treatments (Table 1).

For data confirmation and validation, analyses were performed using different methods, LM and SEM. In the LM assays, the intestinal wall of the larvae increased 2.8-fold (Figure 3a), pupae 2.3-fold (Figure 3b) and adults 3.3-fold (Figure 4c). For the SEM tests, the increases were 2.7-, 2.9- and 2.7-fold, respectively (Figure 3). There was no significant difference between the methods used.

### 3.2. Intestinal Colonization of D. saccharalis by F. verticillioides

We observed that thickening of the intestinal wall, as observed in Figure 2, is mediated by colonization of microvillus intestinal tracts (Figure 4a,d,g,j). No fungal structures were found in the microvilli of insects contaminated with *C. falcatum* (Figure 4b,e,h,k) or not contaminated (Figure 4c,f,i,l), with only gelatinous structures from the artificial diet observed (Figure 4k,l).

We also observed that the fungus established itself in the area, which allowed the formation and germination of reproductive structures among the microvilli (Figure 4g,j).

The combination of LM and SEM also showed increased elongation of the intestinal microvilli in the presence of *F. verticillioides* (Figure 4d). The average microvillus length was 18.66 µm for *F. verticillioides-infected* larvae, whereas for insects contaminated or not contaminated with *C. falcatum*, the average lengths were 11.90 and 10.72 µm, respectively (Figure 5).

## 4. Discussion

The opportunist interaction of borer-feeding insects and microorganisms is well established. Recent studies have started to uncover the interactions between sugarcane borers and their associated fungal microorganisms, namely, *F. verticillioides* and *C. falcatum* [41,42,43], with these interactions including the *F. verticillioides* modulation of insect behavior to cause the insect to act as a vector for the fungus [10].

Here, we provide evidence of how a fungus installs itself and modulates its host’s morphology. The phytopathogenic fungus *F. verticillioides* colonizes the intestinal microvillus region of its host, *D. saccharalis*. In addition, the fungus alters the growth of the insect’s intestinal wall, increasing the intestinal thickness and lengthening the microvilli. Altogether, our data show that these observations are consistent with the “symbiont organism perpetuation hypothesis” [44].

Here, we identified the probable place of “dwelling” of the fungus *F. verticillioides* in its host, a primordial place for the symbiont organisms to be successful in their perpetuation [44]. It has already been shown that this interaction brings benefits to both organisms [45], with no harm to either being identified [5,10], and this installation site proves to be efficient, since it is rich in nutrients and offers the potential for dissemination; thus, it is chosen by several organisms in symbiotic relationships [28,29,44,46].

Several studies have already identified symbiotic microorganisms among the intestinal microvilli of the host [47,48,49,50]; however, these microorganisms are limited to bacteria and viruses, whereas fungal symbionts inhabiting this region have not been found thus far. The microvilli are a strategic place for habitation, since they prevent *F. verticillioides* from coming into contact with digestive enzymes, as the secretion vesicles circulate intact membranes throughout the ectoperitrophic space until they cross the peritrophic membrane. Upon reaching the endoperitrophic space, the vesicles begin to show signs of disintegration [51,52].

Our data are aligned with the results obtained with symbiotic bacteria in ant intestines [47], where the microorganism reproduced and was clearly adapted to the intestinal microvilli region. In addition, there is intensive transport activity in the microvilli, since these are one of the main structures for absorption and enzymatic exchange [53]. This supports the “symbiont organism housing hypothesis” [44], since it would be a place used to initiate targeting of the microorganism to the host’s reproductive system.

We also observed a higher level of complexity in this interaction than has been observed thus far, since this symbiotic relationship between *D. saccharalis* and *F. verticillioides* was studied in a more restricted region. The reason or cause of these modifications is still unknown and is the object of study. Morphological changes in symbiotic organisms are already being observed [54] and are known to impact fitness and thus shape the host phenotype [55,56]. The changes found in the intestinal region generally promote plasticity, elongation, increase or decrease, and/or cell diversification [47,57,58,59,60], as observed from insects to humans, with no specific changes being identified for a given host [54]. The main consequences of intestinal morphological changes described in insects are for metabolic regulation, nutritional balance, pH stabilization and endosymbiotic recycling [47,57,59,60], which may be beneficial for only one symbiotic organism or both.

In our study, we did not identify which consequences the morphological alterations caused and which would be the target of the benefit; however, in similar research [57], it has been speculated that there is a mutual benefit between *D. saccharalis* and *F. verticillioides*, where the insect regulates its metabolism and makes the environment favorable for colonization by the fungus.

Finally, the complexity of the *D. saccharalis*-*F. verticillioides*–sugarcane dynamic may be much more complex and multifaceted than our current understanding of these interactions has revealed. Our results show that, in addition to the fungus manipulating the insect and the plants to promote its own infection and dissemination [10], it may also modulate the insect for its protection and reproduction, just as the insect may modulate the fungus for its metabolic regulation. In this way, our work contributes to elucidating this insect–fungus interaction and may assist in the development of potential tools for controlling these organisms.

## 5. Conclusions

Here, we show that the fungus *F. verticillioides* promotes morphological alteration of the intestinal wall and microvillous structures for its colonization, suggesting that this region may be the host’s gateway to the insect’s reproductive organs.

## Figures and Tables

**Figure 1 pathogens-12-00443-f001:**
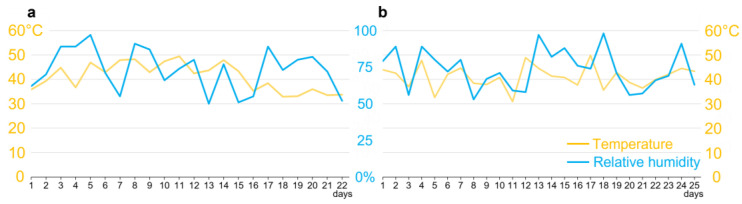
Variation of temperature and humidity in a greenhouse. (**a**) Climatic variation corresponding to the experimental period in sugarcane involving the parental generation of *Diatraea saccharalis*, starting with the first day after inoculation of the plants, with day eight corresponding to the insertion of third-instar caterpillars and between days 20 and 23 to the removal of pupae from the interior of the stems. (**b**) Climatic variation corresponding to the experimental period in sugarcane involving the parental generation of *D. saccharalis*, starting with the insertion of tubes containing eggs in stems, with between days 12 and 15 corresponding to the collection of fifth-instar larvae from stems, and between days 18 and 23 to the collection of pupae.

**Figure 2 pathogens-12-00443-f002:**
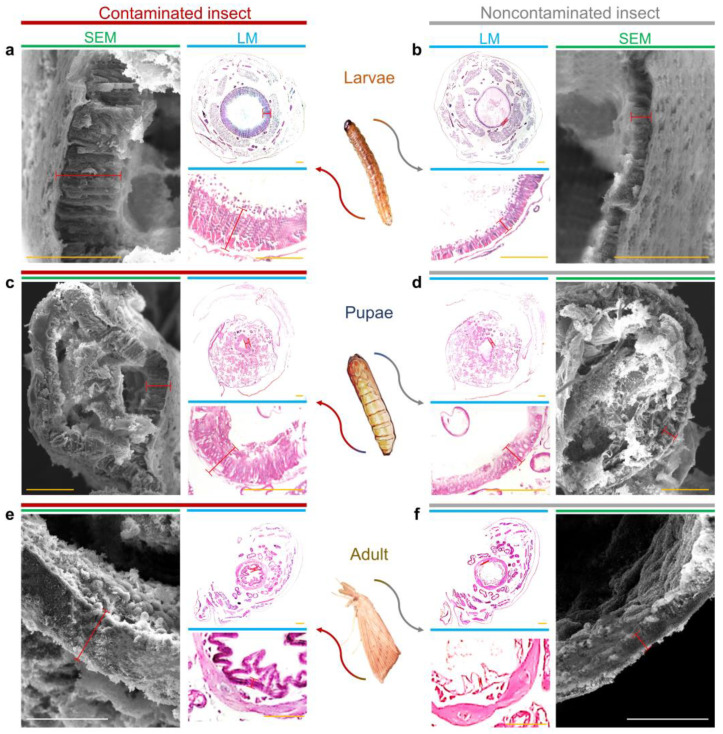
Intestinal wall of *D. saccharalis* by light microscopy (LM) and scanning electron microscopy (SEM). (**a**,**b**) Intestinal wall of the midgut of larvae contaminated and not contaminated with *F. verticillioides*. (**c**,**d**) Intestinal wall of the midgut of pupae contaminated and not contaminated with *F. verticillioides*. (**e**,**f**) Intestinal wall of the midgut of adults contaminated and not contaminated with *F. verticillioides*. Markings in red represent the quantified thicknesses. Yellow bars represent 50 µm, while white bars represent 20 µm.

**Figure 3 pathogens-12-00443-f003:**
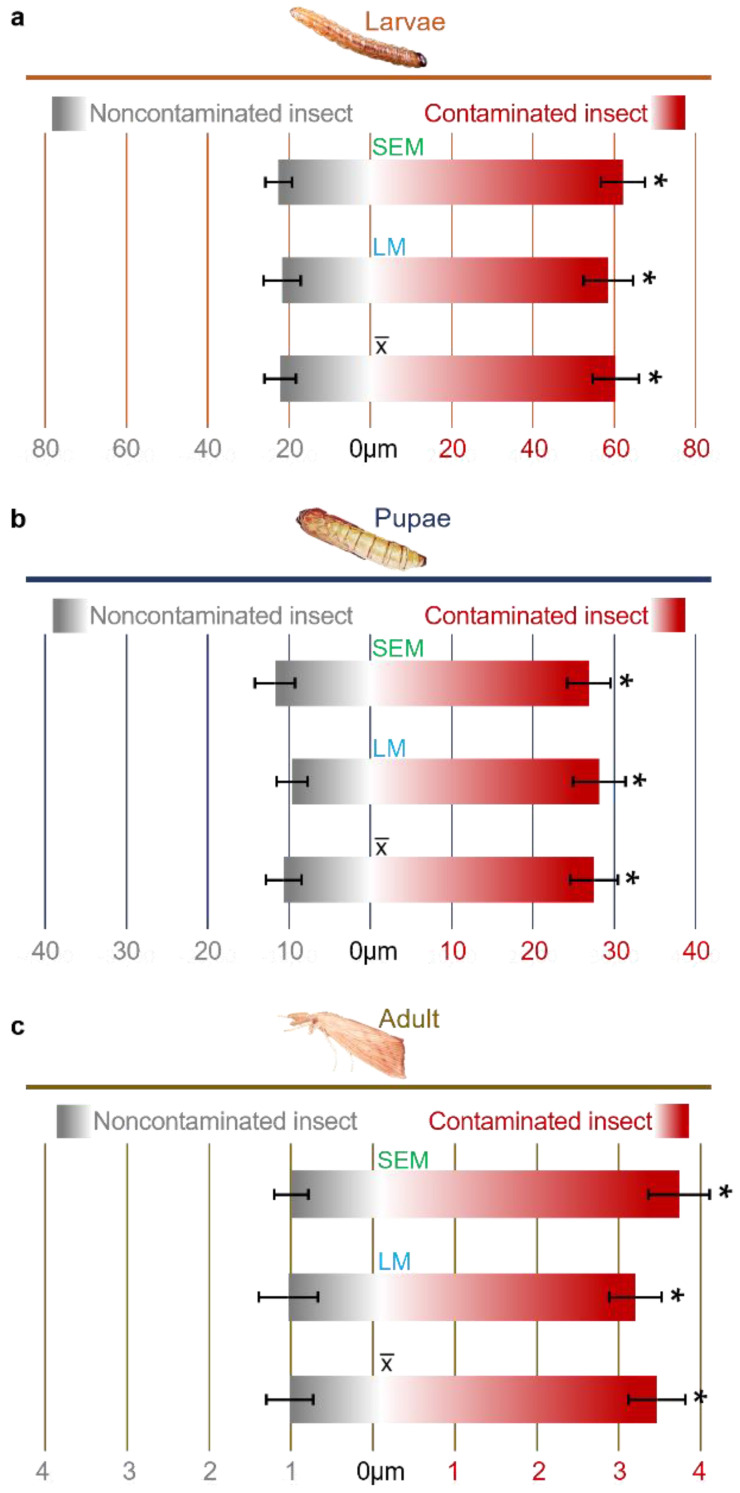
Intestinal thickness of *D. saccharalis* insects contaminated and not contaminated with *F. verticillioides*. (**a**) Thickness of the intestinal wall of larvae measured by scanning electron microscopy (SEM) and light microscopy (LM), also showing the average thickness between the techniques. (**b**) Thickness of the intestinal wall of pupae. (**c**) Thickness of the intestinal wall in adults. Values are the means (±SEs) of ten biological replicates. Asterisks represent significant differences (*t* test, *p* < 0.01) compared to the control.

**Figure 4 pathogens-12-00443-f004:**
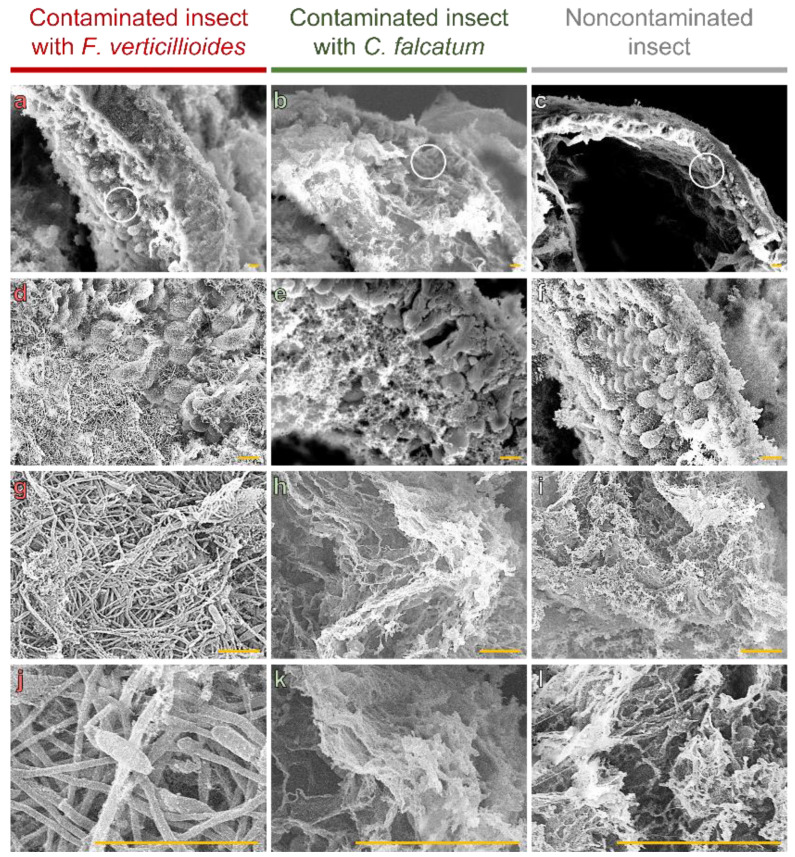
Intestinal microvilli of *D. saccharalis* larvae contaminated or not contaminated with *F. verticillioides* or *C. falcatum* by scanning electron microscopy (SEM). (**a**–**c**) Intestinal surface containing microvilli (example regions are demarcated in white circle), being that of insects contaminated with *F. verticillioides*, (**b**) with *C. falcatum* and (**c**) not contaminated. (**d**) *F. verticillioides* colonizing the spaces between the microvilli. (**e**,**f**) Spaces between empty microvilli or with gelatinous structures from the artificial diet. (**g**) Fungal structures of *F. verticillioides* occupying internal spaces between microvilli, with spore formation and germination. (**h**,**i**) Artificial diet. (**j**) Formation and germination of spores of *F. verticillioides*. (**k**,**l**) Microstructures of the artificial diet. Yellow bars represent 10 µm.

**Figure 5 pathogens-12-00443-f005:**
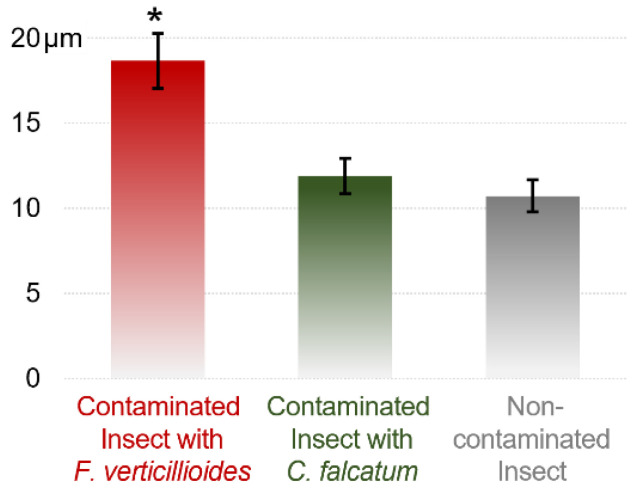
Length of intestinal microvilli of *D. saccharalis* larvae contaminated with *F. verticillioides* or *C. falcatum* or uncontaminated, measured by scanning electron microscopy (SEM). Values are the means (±SEs) of ten biological replicates. Asterisks represent significant differences (*t* test, *p* < 0.05) compared to the control.

**Table 1 pathogens-12-00443-t001:** Intestinal thickness of *D. saccharalis* insects submitted to an artificial diet or sugarcane plants, contaminated or not with fungi, measured by light microscopy (LM). Identical bold letters do not differ significantly between treatments by Tukey’s HSD test.

	Uncontamnated Insect	Contaminated Insect with *C. falcatum*	Contaminated Insect with *F. verticillioides*	*p* Value
*Larvae*				
Artificial Diet	22.57 ± 3.23 **a**	23.97 ± 3.02 **a**	62.23 ± 5.39 **b**	>0.01
Sugarcane Plants	25.07 ± 4.75 **a**	20.03 ± 5.17 **a**	56.53 ± 6.63 **b**	>0.01
*p* value	0.2927	0.1160	0.1085	
*Pupae*				
Artificial Diet	11.73 ± 2.45 **a**	14.47 ± 2.33 **a**	26.90 ± 2.68 **b**	>0.01
Sugarcane Plants	15.17 ± 3.37 **a**	13.90 ± 2.68 **a**	23.47 ± 5.87 **b**	>0.01
*p* value	0.0641	0.5813	0.1736	
*Adult*				
Artificial Diet	1.10 ± 0.24 **a**	1.47 ± 0.24 **a**	3.63 ± 0.37 **b**	>0.01
Sugarcane Plants	1.57 ± 0.41 **a**	1.33 ± 0.33 **a**	3.07 ± 0.75 **b**	>0.01
*p* value	0.0605	0.4518	0.3301

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
