# Peer review of "The Phytopathogen Fusarium verticillioides Modifies the Intestinal Morphology of the Sugarcane Borer"

_pathogens, 2023, doi:10.3390/pathogens12030443_

Round 1

Reviewer 1 Report

The manuscript “Fungal phytopathogen modifies the intestinal morphology of the sugarcane borer” written by Diego et al. described the fungus F. verticillioides alters the intestinal morphology of D. saccharalis. The overall topic is very interesting and some experiments have been performed well. However, some experiments were not well designed, and the data was not well interpreted.

Title: The title can narrow down to specific species of fungal and sugarcane borer

Abstract: line 11 delete “background”, line 16 delete “methods”, line 19 delete “result”.

Introduction:

Is this the first report that fungal can modify the morphology of insects? Otherwise, some related references should be introduced in this section.

Materials and Methods:

Please add the full name of MPBs

Results:

1.     Line195-196: The author said “there was no significant difference between the trials using artificial diet or sugarcane plants, and the only differences were between treatments” but only posted the artificial diet treated SEM and LM pictures, the pictures of sugarcane plants treated also needed. Otherwise, I don’t understand why there has to be two different treatments for this trail.

2.     Fig.2 and Table.1 reflected the same result, the fig.2 included the table.1 data, I don’t think table.1 is needed. Also, the SEM and LM picture of contaminated insect with C. falcatum are needed for comparison with uncontaminated insect and contaminated insect with F. verticillioides

3.     Reslut3.2, Is there any morphology difference for P generation in D. saccharalis or C. falcatum by F. verticillioides compare to noncontaminated insects? Can you observe the intestinal wall of the midgut and intestinal microvilli by SEM and LM?

4.     The increase in midgut thickness is good for the survival of sugarcane borer or not?

References:

Some of the doi were missing in the reference, the author should carefully check the references again.

Author Response

The manuscript “Fungal phytopathogen modifies the intestinal morphology of the sugarcane borer” written by Diego et al. described the fungus F. verticillioides alters the intestinal morphology of D. saccharalis. The overall topic is very interesting and some experiments have been performed well. However, some experiments were not well designed, and the data was not well interpreted.

Thanks for the comment and review. All suggestions are welcome, thus improving our work.

Title: The title can narrow down to specific species of fungal and sugarcane borer

We are grateful for the suggestion, which is the same as Reviewer 3. We have changed the title to include the sxcientific name of the phytopathogen "Fusarium verticillioides".

 Abstract: line 11 delete “background”, line 16 delete “methods”, line 19 delete “result”.

We appreciate the comment, however the insertion of these words is informed in the instructions for submission of manuscripts of the journal Pathogens.

Introduction: Is this the first report that fungal can modify the morphology of insects? Otherwise, some related references should be introduced in this section.

To our knowledge this is the first report dealing with morphological modifications in insect intestines caused by fungal phytopathogens.

Materials and Methods: Please add the full name of MPBs

Thanks for the observation. We corrected it in the text.

Results:

  1. Line195-196: The author said “there was no significant difference between the trials using artificial diet or sugarcane plants, and the only differences were between treatments” but only posted the artificial diet treated SEM and LM pictures, the pictures of sugarcane plants treated also needed. Otherwise, I don’t understand why there has to be two different treatments for this trail.

Thanks for the comment. The two treatments are important to restrict the error caused by the environment and food consumption. There are experiments that we carry out with this interaction in which the behavior of the organisms differs when analyzed in an artificial diet or in sugarcane, in this way, for greater reliability, we carried out both experiments. The average thicknesses obtained by the experiments were entered numerically, with the respective statistical analyzes in Table 1.

  1. 2 and Table.1 reflected the same result, the fig.2 included the table.1 data, I don’t think table.1 is needed. Also, the SEM and LM picture of contaminated insect with C. falcatum are needed for comparison with uncontaminated insect and contaminated insect with F. verticillioides

Thanks for the comment. There are differences between both data, and Table 1 corresponds to the experiment carried out by LM to confirm the data by changing the environment and feeding, as presented in the previous question. In the table we used insects from an artificial diet created in a controlled and controled environment, and from sugarcane plants created in a greenhouse with environmental variations (described in the methodology).

In Figure 2, both SEM and LM were used in insects from an artificial diet, since the table shows that there is no difference when compared with sugarcane.-based diet

Although we could see some similarity between the data, there are different experiments for different purposes, so we believe that they complement each other. This is the reason to keep it in the MS.

  1. 2, Is there any morphology difference for P generation in D. saccharalis or C. falcatum by F. verticillioides compare tononcontaminated insects? Can you observe the intestinal wall of the midgut and intestinal microvilli by SEM and LM?

There is no significant morphological difference, since the P generation used had its first contact with the host in the third instar of its larval stage, that is, its digestive system is already formed and will have only two developmental instars in contact with the fungus. In this way, we used F1 insects from the P generation that have already been exposed to the fungus and probably established the symbiosis. We have already shown (Franco et al 2021 ISME J, cited in the references) that the fungus is present inside the egg. Thus, we ensure that the insect is in contact with its host from moment 0.

In addition, during insect development, the digestive system has its main origin in the pre-hatching phase followed by larval stage (holometabolous insects). In the pupal phase (metamorphosis period) the intestine practically does not present any morphological modification, but a process of "atrophying" (being one of the few organs that remain "intact" in this phase). Then, the adult phase the intestine is smaller and has a funneled digestive system, since the insect does not feed. Although it is not the objective of this work, these steps can be seen in Figure 1.

  1. The increase in midgut thickness is good for the survival of sugarcane borer or not?

Thanks for question. To date, we have no direct answer to this observation. We may infer that this adaptation is somehow helpful to the insect, since its development and survival is not affected by the fungus. A possible benefit of this interaction has been recently reported by our group (please see, Gallan et al. (2023) Sugarcane multitrophic interactions: Integrating belowground and aboveground organisms. Genet Mol Biol 46).”

References: Some of the doi were missing in the reference, the author should carefully check the references again

Thanks for the note. References were checked and missing doi were inserted.

Reviewer 2 Report

The MS presents interesting and potentially important information for ecologists and crop management authorities on symbiotic relationship between fungus and a pest moth. Th MS shows the complex interrelationship between plants and other organisms, including plant pathogens, such as the Fusarium fungus.

While I do not feel qualified to judge the methodical aspects of the MS I have a few general comments on the MS, which might increase its comprehensibility.  

The most interesting to me is the idea that Fusarium may spread and infect the reproductive system of moth. How could this happen? Do the spores transmit through microvilli and penetrate the eggs? Are the eggs infected when layed? Such infections are known from endosymbiotic bacteria in aphids, so could this also happen in this case?

Can fungi be present in the moth feaces? If yes, then would that be how to disseminate and infect the plants or other animals? There are suggestions that gut microbiota may spread from bees to e.g. aphids (Gilliamella in Dysaphis aphids – Kaszyca-Taszakowska & Depa 2022) through feaces or ants. Would that be applicable in case of Fusarium in moth intestine?   

I feel there is suggestion that there may be symbiotic-mutualistic relationship between Diatraea and Fusarium. Please state this more clearly in the MS, if I am right. And perhaps the overgrown microvilli is a result of stress from the negative impact of pathogenic fungus?  

I realise that may questions may be very general, but if that might broaden the range of potential readers, perhaps the Authors might include the answers into the revised version of the MS.  

Also, on Figure 1 - what the bent arrows on left and right of the insect indicate, sections from foregut and hindgut? Because if nothing like that then perhaps they might be removed.     

Author Response

The MS presents interesting and potentially important information for ecologists and crop management authorities on symbiotic relationship between fungus and a pest moth. The MS shows the complex interrelationship between plants and other organisms, including plant pathogens, such as the Fusarium fungus.

While I do not feel qualified to judge the methodical aspects of the MS I have a few general comments on the MS, which might increase its comprehensibility.

Many thanks for the comments. We are grateful for the suggestions to this work and future research.

The most interesting to me is the idea that Fusarium may spread and infect the reproductive system of moth. i) How could this happen? ii) Do the spores transmit through microvilli and penetrate the eggs? iii) Are the eggs infected when layed? iv) Such infections are known from endosymbiotic bacteria in aphids, so could this also happen in this case?

  1. This is really a very interesting point and it is object of our current research. With this paper we provide the starting point: alteration of the intestine morphology and fungal accumulation at the microvilli. Hopefully, we’ll provide this answer in a near future.
  2. The microvilli may be considered as a hotspot of many important physiological activities, such as: vesicle transport, physical-chemical reactions, to mention a few. Therefore, it is tempting to speculate that this region may have a key role in the process.
  • Yes, the eggs are infected and this has been recently shown by our group (please see, Franco et al (2021) ISME J, doi:10.1038/s41396-021-01010-z, cited in the references herein).
  1. Yes, this observation (aphid-bacteria interaction) may occur in other multitrophic interaction, as observed in this work and previous one from our group. 
  2. i) Can fungi be present in the moth feaces? If yes, then would that be how to disseminate and infect the plants or other animals? ii) There are suggestions that gut microbiota may spread from bees to e.g. aphids (Gilliamella in Dysaphis aphids – Kaszyca-Taszakowska & Depa 2022) through feaces or ants. Would that be applicable in case of Fusariumin moth intestine? 
  3. Yes, we have shown a high contamination of the fungi in larval feaces. We have shown (Franco et al 2019) Frontiers in Plant Science 9: 1916) that this is probably the way the the fungus disseminates within the plant tissue. Regarding dissemination, the main route is probably due to eggs, since there are several postures during the adult phase, and the contaminated larvae will spread out throughout the area.
  4. Thanks for the question, we are currently charactering the insect larval microbiota in the presence of the fungus. Preliminary results showed a strong effect on the microbiota composition, bringing an additional level of complexity to this interaction.  

I feel there is suggestion that there may be symbiotic-mutualistic relationship between Diatraea and Fusarium. Please state this more clearly in the MS, if I am right. And perhaps the overgrown microvilli is a result of stress from the negative impact of pathogenic fungus?

Thank you for the comment. We believe that there might be a symbiotic-mutualistic relationship between these organisms. It is clear that this interaction is highly favorable to the fungus, therefore, a well-defined benefit for the insect remains to be established. On the other hand, an interesting study has shown that the presence of F. verticillioides prevents the detection of the borer by C. flavipes, a natural parasitoid, diminishing its biological control efficiency. This is the only benefit for the D. saccharalis identified so far.

We have made changes to clarify this point in MS, thank you.  

I realise that may questions may be very general, but if that might broaden the range of potential readers, perhaps the Authors might include the answers into the revised version of the MS.

Thank you for the comment. We appreciate it and have made some changes according to this suggestion.   

Also, on Figure 1 - what the bent arrows on left and right of the insect indicate, sections from foregut and hindgut? Because if nothing like that then perhaps they might be removed.

No. This shows that on the left is the contaminated insect and, on the right, the non-contaminated one. The arrows have the same color to indicate the respective treatment. Thanks for the note.     

Reviewer 3 Report

The work reported is interesting. However, there are some points that the authors need to address for publication. The points are given below:

1.       In the title, “fungal phytopathogen modifies” should be replaced to “a fungal phytopathogen modifies” or “fungal phytopathogens modify”. Moreover, a title should be general and cover the whole meaning of the study, but in this case, it was too general. At least, the name or the cause of phytopathogen should be mentioned here.

2.       In the abstract, the background is too long, while it should have two sentences at maximum. Moreover, a section for the aim of the study was missing, it should be added in the abstract afterward. The phrase “Since we have previously shown…” was unnecessary, it should be removed. In addition, an overview of methods and procedures used in this study should be shown. At this point, the methodology in the abstract should be rewritten Last but not least, the findings were too little and unhighlighted. Further explanations or research are required.

3.       In the keywords, it should be in the alphabetical order.

4.       In the Introduction:      

a.       Line 38-41, citations are required for this statement.

b.       It took 2 paragraphs to talk about the interactions of plants and pathogens. This is too much, it should be condensed in one paragraph instead, and give more information about the subjects and the needs of this study.

c.       Line 56-57, in this study, the fungi should be emphasized, not bacteria. This had better replace by literatures about intestines and fungi.

d.       Line 66-68, if there are any literatures about informations obtained for the fungal colonization, they should be added.

e.       Line 68-75, why the results were mentioned here? At this point, it should be the objectives of the study. This point needs to be seriously rewritten.

5.       In the materials and methods:

a.       How long was the insects kept? What are their characteristics, e.g., life-span, colors, size, etc. Please specify.

b.       Although the light, temperature and humidity of the sugarcane greenhouse were natural, they should be measured in details.

c.       If there was a control treatment, the number of replications should be mentioned.

d.       Line 99-100, why were nipagin and formaldehyde removed?

e.       When plants were inoculated with pathogens, it should have been experimental designs and distances between them. If not, the cross transmission between plants could cause a mess.

f.        For how long were the insects infected with the fungi?

g.       In my of opinion, HOCl is a antifungal chemical, the colonized fungi may have been washed from the larvae.

h.       Line 147, how to distinguish the contaminated and uncontaminated insects, while they had been infected by the fungi?

i.        The manufacturer of the microtome should be added.

j.        The procedure how the insects’ intestine was collected to be observed under the microscope.

6.       Results:

-Lines 180-182: move to methods

-Line 213-215: move to methods

7.       In the discussion: line 262-265, this statement needs a literature.

8.       Lack of conclusion section

Author Response

The work reported is interesting. However, there are some points that the authors need to address for publication. The points are given below:

  1. In the title, “fungal phytopathogen modifies” should be replaced to “a fungal phytopathogen modifies” or “fungal phytopathogens modify”. Moreover, a title should be general and cover the whole meaning of the study, but in this case, it was too general. At least, the name or the cause of phytopathogen should be mentioned here.

Thanks for the comment, which is similar to Reviewer 1. We replaced the term "modifies" with "modify". In addition, we have included the scientific name of the pathogen "Fusarium verticillioides".

  1. In the abstract, i) the background is too long, while it should have two sentences at maximum. ii) Moreover, a section for the aim of the study was missing, it should be added in the abstract afterward. iii) The phrase “Since we have previously shown…” was unnecessary, it should be removed. iv) In addition, an overview of methods and procedures used in this study should be shown. At this point, the methodology in the abstract should be rewritten. v) Last but not least, the findings were too little and unhighlighted. Further explanations or research are required.
  2. Thanks for the suggestion. We synthesized the background from 80 words to 50.
  3. Thanks for the observation. The objective was inserted at the end of the "background" section, as guided by the magazine.
  • Thanks for the suggestion, the sentence has been removed.
  1. Thank you for your comment, the methodology of the abstract has been rewritten for a better approach to the methods and procedures used.
  2. Thank you for your comment, the results section of the summary has been changed, addressing more data for greater understanding, and highlighting.
  3. In the keywords, it should be in the alphabetical order.

Thanks for the note. Keywords have been placed in alphabetical order.

  1. In the Introduction:       
  2. Line 38-41, citations are required for this statement.

Thanks for the comment, the quote has been inserted.

  1. It took 2 paragraphs to talk about the interactions of plants and pathogens. This is too much, it should be condensed in one paragraph instead, and give more information about the subjects and the needs of this study.

Thanks for the suggestion, the paragraphs have been condensed into just one, and the information has been better elaborated.

  1. Line 56-57, in this study, the fungi should be emphasized, not bacteria. This had better replace by literatures about intestines and fungi.

Thanks for the suggestion. The paragraph has been reformulated and data on interactions involving colonization of insect intestines by fungi has been inserted.

  1. Line 66-68, if there are any literatures about informations obtained for the fungal colonization, they should be added.

In the context of the interaction presented, we could not find any study in the literature describing the mechanism of fungal colonization of the intestine linked to the transport of the host to the reproductive organ. Furthermore, we are currently working to unravel this highly complex mechanism.

  1. Line 68-75, why the results were mentioned here? At this point, it should be the objectives of the study. This point needs to be seriously rewritten.

Thanks for the comment. The results obtained were withdrawn from this point and the paragraph was reformulated.

  1. In the materials and methods:
  2. How long was the insects kept? What are their characteristics, e.g., life-span, colors, size, etc. Please specify.

The insects were maintained on an artificial diet throughout their larval stage. The information was reformulated in the text for better understanding and inserted literature featuring the insect. Thanks for the suggestions.

  1. Although the light, temperature and humidity of the sugarcane greenhouse were natural, they should be measured in details.

Thanks for the comment. Temperature and humidity were measured during the experiment and inserted into the MS.

  1. If there was a control treatment, the number of replications should be mentioned.

Thanks for the comment.Ssection 2.9 was changed including the repetitions and section 2.10 was created by inserting corresponding data.

  1. Line 99-100, why were nipagin and formaldehyde removed?

These compounds are used artificial diets of various insects to avoid fungal contamination, therefore, they were removed in order to allow fungal growth in the diet.

  1. When plants were inoculated with pathogens, it should have been experimental designs and distances between them. If not, the cross transmission between plants could cause a mess.

Thank you for the observation. Yes, we took the necessary measures to prevent cross-transmission, taking into account the spacing between treatments. At the end of the experiment we collect material for molecular analysis of the plants in order to validate the results that show only the inoculated organism.

  1. For how long were the insects infected with the fungi?

Third-instae insect larvae from parental generation were exposed to fungal-infected diet to the pupal stage following external hygiene steps. As the microorganism presents vertical transfer by the vector, it remains in the insect organism throughout its life cycle cicle and offspring. Data are described in section 2.6.

  1. In my of opinion, HOCl is a antifungal chemical, the colonized fungi may have been washed from the larvae.

Thanks for the comment. Yes, both NaClO and CuSO4 used for cleaning have antifungal activity. However, both compounds were not used in larval stage, only in pupae and eggs promoting external fungal sterilization, and having no internal action. This protocol has been described by our group in (Franco et al (2021) ISME Journal, doi:10.1038/s41396-021-01010-z).

  1. Line 147, how to distinguish the contaminated and uncontaminated insects, while they had been infected by the fungi?

The experimental procedures were done separately, in Petri dishes, glass tubes or screened cages, with no contact between treatments.

  1. The manufacturer of the microtome should be added.

The equipment data are already described in section 2.7 ("The samples were sectioned using a Leica RM 2255 rotational microtome with a thick-ness of 8 μm.").

  1. The procedure how the insects’ intestine was collected to be observed under the microscope.

Thanks for the comment. The synthesized procedure is described in section 2.8, but we inserted the corresponding literature where it better describes the process.

  1. Results:

-Lines 180-182: move to methods

Thanks for the suggestion. The presented content was taken from the results and incorporated into the methodology.

-Line 213-215: move to methods

Thanks for the suggestion. The presented content was taken from the results and incorporated into the methodology.

  1. In the discussion: line 262-265, this statement needs a literature.

Thanks for the comment. Literature was inserted.

  1. Lack of conclusion section

The conclusion section is designated as an optional research MS, as described in the Pathogens journal instructions. However, as mentioned, we also found this section important and included it into the MS. Thank you.

Round 2

Reviewer 3 Report

Agree